# The need for patient-centric medicine design: investigating key physical characteristics of oral solid medications to improve acceptance of older patients in Addis Ababa, Ethiopia

**Mohammed Chane Assefa**[1,2*], **Getahun Paulos**[3], **Fikadu Ejeta**[2], **Dereje Kebebe Borga**[4*]

**1** School of pharmacy, College of Medicine and Health Science, University of Gondar, Gondar, Ethiopia, **2** School of Pharmacy, Faculty of Health Science, Jimma University, Jimma, Ethiopia, **3** Department of Pharmacy, Komar University of Science and Technology, Iraq, **4** Department of Pharmacy, College of Applied Natural sciences, Adama Sciences and Technology University, Adama, Oromia, Ethiopia

* mohammedchane0119@gmail.com (MCA), dereje.keborg@gmail.com (DKB)

## Abstract

Patient-centric medicine design emphasizes developing medication products tailored to patients' specific needs and preferences, including size, shape, color, texture, packaging, and labeling. However, despite increasing recognition of patient-centered pharmaceutical design, evidence remains scarce on how medication and patient characteristics influence medication acceptance among older patients, particularly in low-resource settings such as Ethiopia. This study aimed to identify factors associated with the acceptance of oral solid medications among older adults. An institutional-based cross-sectional study was conducted from May to July 2024 in five public hospitals in Addis Ababa, Ethiopia. Data were collected through an interviewer-administered questionnaire capturing patient-related information (socio-demographics, clinical conditions, and medication beliefs) and medication-related characteristics (size, shape, score line, packaging, texture, dosage form, and labeling). Medication-level acceptance was measured using the MAQ-2019 tool. Descriptive statistics summarized the data, and bivariable and multivariable logistic regression analyses identified predictors of acceptance. Within-patient clustering was accounted for using cluster-robust standard errors. Among 408 older patients and 1,256 oral solid medications, 75% of medications were accepted. Medications sized 6–9 mm were more likely to be accepted (AOR = 6.50, 95% CI: 3.84–10.99, p < 0.001), while those ≥18 mm were less likely to be accepted (AOR = 0.15, 95% CI: 0.06–0.35, p < 0.001) compared with ≤6 mm. Medications with a score line were twice as likely to be accepted (AOR = 1.99, 95% CI: 1.32–3.00, p = 0.001). Older patients preferred small to medium-sized, coated, and scored tablets. Findings underscore the importance of age-appropriate formulation design and patient involvement in medication development.

**Data availability statement:** All relevant data are within the manuscript and its Supporting Information files.

**Funding:** The author(s) received no specific funding for this work.

**Competing interests:** The authors have declared that no competing interests exist.

**Abbreviation:** MAQ, Medication Acceptability Questionnaire; OSM, Oral Solid Medications; PCMD, Patient-Centric Medicine Design; WHO, World Health Organization; SPC, Summary of Product Characteristics

## Introduction

Healthcare systems are facing more and more difficulties as a result of the world's ageing population, especially with regard to the safe and efficient administration of medication [1]. Over 962 million people worldwide were 60 years of age or older as of 2020; by 2050, that number is expected to rise to over 2.1 billion [2,3]. Similar demographic shifts are taking place in Ethiopia. There will be a substantial change in the nation's healthcare requirements as the number of people 60 and older is predicted to increase from an estimated 5.3 million in 2019 to 14.8 million by 2050 [4]. Age-related chronic conditions like cancer, diabetes, and cardiovascular diseases are becoming more common along with this demographic trend; many of these conditions are primarily treated with long-term use of several medications (polypharmacy), the majority of which are given in oral solid dosage forms (OSDFs) [5]. Even though these dosage forms have advantages like accuracy, stability, and storage convenience, they frequently fall short of meeting the unique physical, mental, and cognitive requirements of older patients. An older person's ability and willingness to correctly use standard medication forms can be significantly impacted by conditions like arthritis, reduced manual dexterity, dysphagia, visual impairment, and cognitive decline [6]. As a result, inappropriate medication modification practices such as tablet splitting, crushing, or dose omission may occur, potentially compromising treatment outcomes.

In this context, the concept of patient-centered medicine design (PCMD) has gained increasing importance. PCMD promotes the development of pharmaceutical products that take into account the physiological characteristics, functional abilities, and sociocultural context of specific patient populations, particularly vulnerable groups such as the older adults [7]. According to the European Medicines Agency (EMA), medication acceptability is defined as the overall ability and willingness of a patient to take a medicine as intended, or of a caregiver to administer it as prescribed [8]. Acceptability is a multidimensional concept encompassing several interrelated aspects, including swallow ability, palatability (taste and aftertaste), handling characteristics (such as size, shape, weight, and ease of manipulation), visual appearance (color and distinguishability), and the need for medication modification [9].

In the present study, acceptability was assessed through patient-reported experiences focusing specifically on physical formulation-related characteristics of OSDFs, including tablet or capsule size, shape, surface texture or coating, color, taste, labeling, ease of handling, and the need to alter the dosage form (e.g., breaking or crushing). These aspects were evaluated because they are directly linked to daily medication use and are particularly relevant for older patients with functional limitations [10].

Numerous international studies have reaffirmed how physical characteristics of oral solid medications affect people's acceptance of their medications. Oval tablets weighing about 250 mg, for instance, were much easier for older patients to swallow than larger or rounder tablets, according to a randomized study conducted in Germany [5]. Comparably, qualitative research conducted in the UK and Japan revealed that shape and visual appeal had a major impact on older patients' acceptance of oral solid medications, particularly for those who have swallowing issues [10]. It has also

been demonstrated that surface texture and coating enhance palatability and ease of swallowing; tablets with coatings are perceived as smoother, less bitter, and more palatable than those without coatings [11].

Medication recognition and acceptance have been found to be significantly influenced by appearance, including color and distinguishability, in addition to dimensional features, especially for older patients who are dealing with polypharmacy [12]. In addition to helping patients with visual or cognitive impairments avoid confusion, bright, distinct colors aid in the differentiation of medications [10]. Preferences, however, can differ depending on patient experience and culture. Although some European studies indicated that white-colored tablets were the most popular, while, other studies highlighted the benefits of dual-colored medications to make identification easier [13].

Often undervalued, palatability has been repeatedly associated with acceptance and appropriate use. Taste, coating, and aftertaste all had a significant impact on whether or not older patients felt comfortable taking their medications, according to UK research. It was more likely that tablets with palatable taste profiles and smooth, slick coatings would be taken regularly and unaltered. On the other hand, many patients changed or skipped doses due to rough textures or disagreeable tastes, increasing the chance of treatment failure or unfavorable results [10,14].

Despite this growing body of evidence, most available data originate from high-income countries with well-established regulatory frameworks, strong patient involvement in medicine design, and greater access to age-appropriate formulations. In contrast, evidence from low-resource settings such as Ethiopia remains scarce. Differences in healthcare infrastructure, regulatory enforcement, availability of dosage form options, patient education, health literacy, cultural perceptions of medicines, and reliance on generic products may substantially influence how older patients perceive and use oral medications. In Ethiopia, the pharmaceutical market is dominated by imported or locally manufactured generic medicines, which are rarely tailored to older patients' functional needs. Additionally, limited patient counseling, lower awareness of medication-related risks, and economic constraints may further affect acceptability and medication use behaviors [5,10].

Given Ethiopia's ongoing demographic and epidemiological transition, understanding context-specific factors influencing medication acceptability among older adults is essential. Incorporating older patients' perspectives into medicine selection, procurement, and formulation decisions may help reduce inappropriate medication modification, improve adherence, and enhance therapeutic outcomes. Therefore, this study aims to investigate physical formulation-related factors influencing the acceptability of oral solid dosage forms among older patients in Addis Ababa, Ethiopia. The findings are expected to provide locally relevant evidence to support patient-centered pharmaceutical care and inform future policy and practice in similar low-resource settings.

## Method and material

### Study settings

This study was conducted in selected public tertiary and general hospitals in Addis Ababa City Administration, Ethiopia. These hospitals provide a full range of outpatient and inpatient services, including internal medicine, cardiology, endocrinology, oncology, infectious diseases, surgery, and geriatrics-related care. They are not specialized in a single indication, but rather function as major referral and teaching hospitals serving patients with diverse acute and chronic conditions from Addis Ababa and other regions of the country.

Addis Ababa has the most developed health infrastructure in Ethiopia, accounting for approximately 21% of the country's healthcare facilities, and employs the highest number of qualified healthcare professionals. The city has 12 public hospitals (seven administered by the Addis Ababa Health Bureau, four by the Federal Ministry of Health, and one by Addis Ababa University), as well as more than 34 private hospitals. It is estimated that 20,000–21,000 healthcare professionals work in Addis Ababa, of whom physicians constitute approximately 14% and nurses 47% [15]. Due to the availability of advanced diagnostic and treatment services, older patients with a wide range of chronic diseases commonly seek care at these hospitals.

## Study design and period

An institutional-based cross-sectional study was conducted from May to July 2024 to examine the association between physical formulation characteristics of oral solid dosage forms (OSDFs) and medication acceptability among older patients.

## Populations

**Source population.** All older patients aged 60 years and above attending outpatient departments of public hospitals in Addis Ababa.

**Study population.** Older patients aged ≥60 years who attended outpatient services at the five selected public hospitals during the data collection period and met the eligibility criteria.

**Eligibility criteria. Inclusion criteria:** Older patients aged 60 years or older who was attending outpatient clinics had been chronically using at least one prescribed oral solid medication for ≥3 months, were able to present at least one of their currently used oral solid medications at the time of the interview, and voluntarily agreed to participate.
**Exclusion criteria:** Older patients who were unable to provide reliable and valid information due to terminal illness, severe cognitive impairment (e.g., advanced dementia), or any physical condition that hinders their ability to give informed consent or participate in interviews were excluded.

## Sampling

**Sample size determination.** The sample size was calculated using the single population proportion formula, assuming a 50% expected acceptance level of oral solid medications among older patients, a 5% margin of error, and a 95% confidence level (Z = 1.96):

$$n = \frac{(Z_{a/2})^2 \times P(1-P)}{d^2}$$

Where: n = required sample size
Z = 1.96
P = 0.5
d = 0.05

This yielded a sample size of 384. After adding a 10% non-response allowance, the final sample size was 422 participants.

## Sampling technique

Five public hospitals; Tikur Anbessa Specialized Hospital, St. Paul's Hospital Millennium Medical College, Zewditu Memorial Hospital, Menelik II Referral Hospital, and Yekatit 12 Hospital were purposively selected because they are major referral hospitals serving large numbers of older patients with chronic diseases.

Based on hospital records, approximately 20,100 older patients attend outpatient clinics monthly across these hospitals. The total sample size was proportionally allocated to each hospital using:

$$n_i = \frac{N_i \times n}{N}$$

Where **ni** = older patients required sample size; **Ni** = older patient source sample size; **n** = total sample size; **N** = total population.

Resulting in allocations of 126, 110, 69, 61, and 42 participants, respectively.

Within each hospital, systematic random sampling was applied. Eligible patients were selected at regular intervals from outpatient attendance lists after completion of clinical consultation and medication dispensing, ensuring that participants could present their medications during data collection.

## Data collection methods and procedure

Data were collected using a semi-structured, interviewer-administered questionnaire designed to assess formulation-related acceptability of OSDFs. The questionnaire included sections on: Socio-demographic characteristics, Aging- and disease-related factors, Behavioral and belief-related factors, and Physical characteristics of oral solid medications, including size, shape, color, surface texture/coating, taste, score line, packaging, labeling readability, and need for modification. The questionnaire was adapted from validated instruments used in previous studies [10,16,17].

The original questionnaire was developed in English, translated into Amharic, and then back-translated into English by senior academic staff who were not aware of the original questionnaire, to ensure semantic consistency. Discrepancies were resolved through consensus among the research team and language experts. Data were collected by three Bachelor of Pharmacy professionals trained in standardized interviewing techniques.

## Assessment of oral solid medications

Information on medications was obtained through: Physical inspection of medicines presented by participants, and review of available Summary of Product Characteristics (SmPCs). As tablet and capsule dimensions are not routinely included in SmPCs, tablet size was measured directly using a calibrated tablet caliper (Madukani 2009, GGG-C 111B, India).

For each participant, up to five oral solid medications were assessed. This approach was used because prior evidence indicates that evaluating up to five medications captures more than 90% of chronically used medicines among older patients, while minimizing respondent burden [18].

Measurement of medication acceptability

Medication acceptability was measured using the Medication Acceptance Questionnaire 2019 (MAQ-2019). The questionnaire evaluates acceptability through multiple dimensions, including: ease of swallowing, taste and aftertaste, handling and manipulation,visual appearance and identification, and willingness to take the medication without modification.

Each item was rated on a Likert scale. A composite acceptability score was calculated for each medication. Scores <66 indicated poor acceptability, while scores ≥66 indicated good acceptability [16].

## Variables of the study

**Independent variable. Socio-demographic factors:** age, sex, education, income, residence, marital status, occupation, family support, and health insurance.

**Aging- and disease-related factors**: type and duration of illness, comorbidities, severity of condition, frequency of healthcare visits.

**Behavioral and belief-related factors**: medication beliefs (necessity, concern, general harm), knowledge, lifestyle behaviors.

**Medication-related factors**: size, shape, color, coating/texture, taste, score line, packaging, dosage form, labeling readability.

**Dependent variable.** Acceptability of oral solid medications among older patients.

## Data quality management

To ensure the quality of the data, the following activities were undertaken: careful design of the questionnaire, training of data collectors, close supervision of the data collection procedures, proper categorization and coding of the data, reviewing the collected data for accuracy and completeness by data collectors, and checking the recorded data. A pretest of the

questionnaire was also conducted on 5% of the total sample size before the actual data collection, and necessary amendments were made based on the feedback. The participants in the pretest were excluded from the final study. The internal consistency of the acceptance questionnaires was validated and had a Cronbach's alpha of 0.77.

## Data entry, analysis and model building

After data collection, each questionnaire was checked for completeness and consistency by the data collector. The collected data was entered and analyzed using Statistical Package for Social Science (SPSS) version 27. Descriptive statistics were used to summarize the demographic characteristics of the participants and the distribution of responses to the survey questions by using tables, graphs, and text. Inferential statistics, such as binary logistic regression, were employed to examine the associations between formulation characteristics and medication acceptance among the older population, and a P value of <0.05 was used to declare statistical significance at a 95% confidence interval (CI).

We examined factors associated with the acceptance of oral solid medications at the medication level (accepted vs. not accepted, defined using the MAQ-2019 threshold ≥66 with necessary modification) among 1,256 medications nested within 408 patients. Candidate predictors were selected from all measured variable using bivariable screening (p < 0.20). The final set included education level; number of oral solid medications [1–5]; visual impairment; hypertension; diabetes; cardiac-related disorders; allergy/inflammation; infection/prophylaxis; overall medical-condition severity; frequency of hospital visits; medication-related beliefs (general harm, necessity, concern, knowledge); and medication design features (size, shape, score line, packaging, texture, dosage form, and labeling readability).

Crude odds ratios (CORs) were estimated using bivariable logistic regression. Variables meeting the inclusion criterion (p < 0.20) or considered clinically important were entered simultaneously into a multivariable logistic regression to obtain adjusted odds ratios (AORs) with 95% confidence intervals (CIs) and p-values. Reference categories were defined to align with Table 6 (e.g., college and above education; one medication; no visual impairment; no hypertension; no diabetes; no cardiac disorder; no allergy/inflammation; no infection/prophylaxis; severe condition; monthly visits; weak general-harm belief; strong necessity belief; weak concern belief; good knowledge; size ≤6 mm; round shape; no score line; blister packaging; coated/smooth texture; tablet dosage form; easy-to-understand labeling).

Because multiple medications were contributed per patient (mean 3.1), within-patient correlation was accounted for using cluster-robust standard errors at the patient level. Model diagnostics included assessments of multi-collinearity (variance inflation factor <10 acceptable), calibration (Hosmer–Lemeshow goodness-of-fit test), discrimination (area under the receiver operating characteristic curve, AUC), and influence (leverage and delta-beta). Missing data were handled by complete-case analysis, as the overall proportion of missing was < 5%. All analyses were conducted in SPSS version 27, using two-sided tests with α = 0.05. Variables labeled "adjusted" in Table 6 correspond to this multivariable model.

## Operational definition of terms

**Older people:** Participants in Ethiopia who are aged 60 years or older, exhibiting age-related physical, social, or psychological characteristics that may influence their medication use

**Key characteristics of oral solid medications**: Refers to the specific attributes and properties of medications that are formulated in solid form and intended for oral administration, such as size, shape, color, palatability, score line, packaging, and others.

**Patient Acceptance**: An older participant willingly and effectively takes the prescribed oral solid medication as instructed, with neutral or positive attitudes towards its characteristics. The study participants' oral solid medication acceptance was assessed using a five-item Likert scale with twenty-two statements from the Medicine Acceptance Questionnaire 2019 (MCQ) with some modifications fit for this study. Participants with a score of less than 50% were considered to have poor acceptance, while those with a score of 50% and above were considered to have good acceptance [16].

**Patient-centric:** Designing oral solid medications with **features that address common challenges faced by older users**, such as handling, swallowing, identification, and memorability.

**Size:** The largest dimension of the oral solid medications refers to the length of oval or oblong tablets and capsules or the diameter of round tablets.

**Solid oral medications:** Pharmaceutical formulations that are in a solid state, typically in the form of tablets, capsules, or powders.

**Summaries of Product Characteristics (SPCs):** A description of a medicinal product's properties and the conditions attached to its use. It explains how to use and prescribe a medicine. It is used by healthcare professionals, such as prescribers, nurses, and pharmacists.

### Ethical consideration

Ethical clearance for the study was obtained from the Institutional Review Board (IRB) of the Institute of Health, Jimma University (Ref. No: JUIH/IRB176/24). The IRB approved the use of verbal informed consent due to the minimal risk nature of the study and the context of data collection. Verbal consent was obtained from each participant after providing a clear explanation of the study's purpose, procedures, benefits, and risks. The data collectors documented the consent process by recording participants' agreement on a standardized consent checklist. Each verbal consent was witnessed by a third party such as health care providers, caregivers and other community members, and this process was conducted in accordance with the IRB's approved ethical protocol. Participation was entirely voluntary, and all participants were informed of their right to decline or withdraw from the study at any time. To ensure confidentiality, no personal identifiers were collected; anonymous questionnaires were used, and all data were securely stored.

## Results and discussion

### Result

**Socio-demographic characteristics and respective OSM use.** A total of 408 older patients (97% response rate) were enrolled in the study, and a total of 1256 oral solid medications were taken by those older patients, with 3.1 oral solid medications for each patient on average. The mean age of the participants was 68.1 (SD ± 7) with ages ranging from 60 to 91 years. More than two-thirds of them (69.6%) were found to be less than 70 years old, and 66.7% of oral solid medications were taken by those groups of patients (Fig 1).

The gender distribution was nearly equal, with 54.4% male and 56.1% oral solid medication consumption. Of the total older patients, almost half (44.9%) identified as Orthodox Christian, followed by Muslims (36.5%) in religion, with 44.8% and 36.1% of oral solid medication consumption, respectively. The majority (78.2%) of participants had a support from family or caregivers and those older patients consumed 78.9% oral solid medications. Geographically, the study population originated primarily from Addis Ababa (34.1%) and Oromia (28.7%) with 32.6% and 28.7% of oral solid medication use respectively. (Table 1).

**Personal and therapy-related characteristics and respective OSM use.** Regarding characteristics of the older patient population in the study, majority the participants (74.5%) were not alcohol drinkers with 74.7% of oral solid medication consumption. More than half (52.9%) of the participants were not using any dosing reminders, while 27% were relying on family members to help them remember to take their medications with 49.8% and 30% of oral solid medication consumption respectively. The majority of patients (31.4%) were taking 3 medications (30.1%), followed by 29.4% taking 2 medications (19.1%) with mean (±SD) of 3.1 (±1) (Table 2).

**Aging-related changes/clinical characteristics and respective OSM use.** Regarding age-related problems, visual impairment is the most prevalent, affecting 28.7% of the participants, followed by swallowing difficulties (27.7%) and oral solid medications, with 32.4% and 32.1% of oral solid medication consumption, respectively. Among the

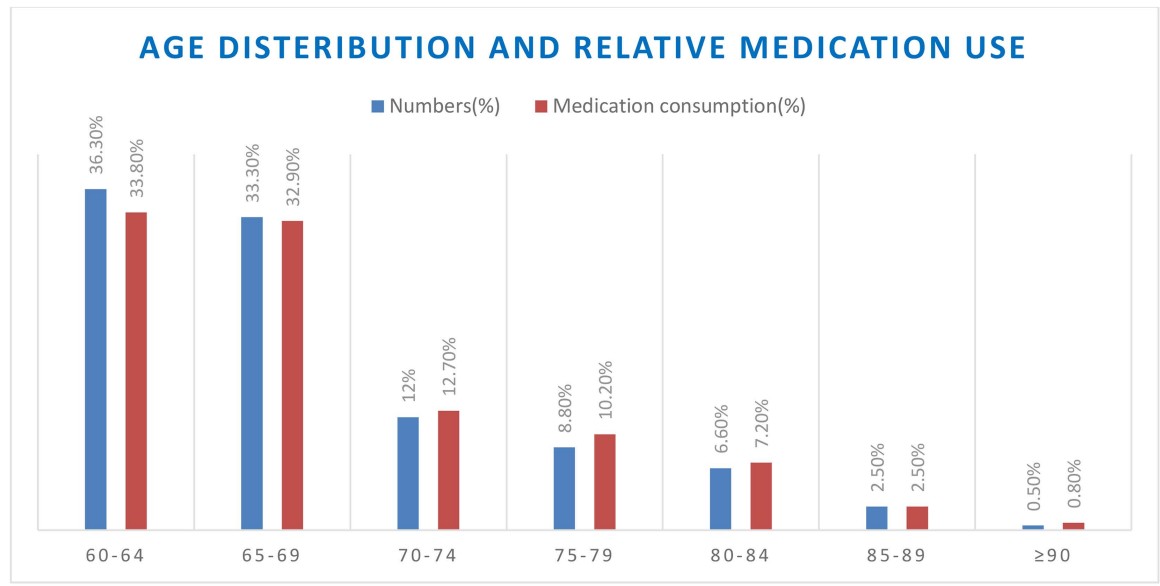

**Fig 1. Age distribution and corresponding oral solid medication consumption of the study participants (older patients).** For each 5-year age category (60–64, 65–69, …,≥90), blue bars indicate the percentage of participants in that category and red bars indicate the percentage of total OSM consumed by that category (n = 408). Age and current OSM were obtained by interview and observation during May to July 2024 at five selected hospitals, Addis Ababa, Ethiopia. Values are proportions; no error bars are shown. Abbreviation: OSM, oral solid medication.

disease conditions affecting older patients, hypertension emerged as the most common (43.1%) with 50.2% of oral solid medication usage, followed by other cardiac-related disorders and diabetes mellitus. (Table 3).

**Behavioral variables and respective OSM use.** This study examines the beliefs of older patients regarding their medications, focusing on personal, religious, cultural, and knowledge-based perspectives. Regarding personal beliefs, the majority of respondents (63.7%) identified as "weak believers" regarding overuse of medications, indicating a better trust in oral solid medications and their health care providers, with an average score of 2.6 on a five-point scale. On the necessity of medications, more than two-thirds of respondents (67.6%) were "strong believers," with an average score of 3.8 (Table 4).

**Characteristics of oral solid medications.** A total of 1,256 oral solid medications were evaluated, with an average consumption of 3.1 medications per patient. Among these medications, the majority (36%) fell within the 6–9 mm size range. This was followed by 17.8% of medications in the 9–12 mm category and 17.5% in the size range of 6 mm or smaller (Fig 2).

The predominant (73.2%) oral solid dosage medications were round. The majority of the oral solid medicines were white or off-white (59.2%), followed by yellow (13.1%). The taste profile of the oral solid medicines showed that most had a bitter taste (57.8%), while 19.5% were neutral, and the majority of the oral solid medicines (59.6%) had a single score line, followed by none score lined (35%). The texture analysis revealed that a majority of the oral solid medicines (61.1%) were uncoated (rough), while 37.1% were coated (smooth), and the majority of the oral solid medicines were in tablet form (91.9%), with only 8.1% being capsules (Table 5).

**Acceptances of oral solid medications among study participants.** From a total of 1,256 oral solid medications evaluated in this study, 945 medications, which correspond to 75%, were accepted by older patients (Fig 3).

**Factors associated with acceptance of oral solid medications of older patients.** Both bivariable and multivariable logistic regression analyses were done to identify factors associated with the medication acceptance of older patients.

**Table 1. Distributions of socio-demographic variables and corresponding oral solid medication consumption of the study participants (older patients) in Addis Ababa, Ethiopia.**

| Sociodemographic information | | Frequency | Percent | respective OSM consumption | Percent |
|---|---|---|---|---|---|
| Sex | Male | 222 | 54.4 | 705 | 56.1 |
| | Female | 186 | 45.6 | 551 | 43.9 |
| Religious affiliation | Orthodox | 183 | 44.9 | 563 | 44.8 |
| | Muslim | 149 | 36.5 | 453 | 36.1 |
| | Protestant | 66 | 16.2 | 207 | 16.5 |
| | Others * | 10 | 2.5 | 33 | 2.6 |
| Residence | Urban | 311 | 76.2 | 949 | 75.6 |
| | Rural | 97 | 23.8 | 307 | 24.4 |
| Marital status | Married | 214 | 52.5 | 638 | 50.8 |
| | Widowed | 131 | 32.1 | 428 | 34.1 |
| | Divorced | 35 | 8.6 | 104 | 8.3 |
| | Separated | 24 | 5.9 | 73 | 5.8 |
| | Single | 4 | 1 | 13 | 1 |
| Region | Addis Ababa | 139 | 34.1 | 410 | 32.6 |
| | Oromia | 117 | 28.7 | 361 | 28.7 |
| | Southern regions | 60 | 14.7 | 207 | 16.5 |
| | Amhara | 55 | 13.5 | 158 | 12.6 |
| | Tigray | 20 | 4.9 | 59 | 4.7 |
| | Others ** | 17 | 4.3 | 61 | 4.9 |
| Level of education | Illiterate | 223 | 54.7 | 697 | 55.5 |
| | Primary school | 66 | 16.2 | 202 | 16.1 |
| | Secondary school | 62 | 15.2 | 192 | 15.3 |
| | College and above | 57 | 13.9 | 165 | 13.1 |
| Occupation | Un employed | 177 | 43.4 | 559 | 44.5 |
| | Private work | 149 | 36.5 | 431 | 34.3 |
| | Retired | 52 | 12.7 | 165 | 13.1 |
| | Farmer | 27 | 6.7 | 95 | 7.6 |
| | Other *** | 3 | 0.7 | 6 | 0.5 |
| Average monthly income | ≤2000 | 158 | 38.7 | 499 | 39.7 |
| | 2001-4999 | 166 | 40.7 | 490 | 39 |
| | ≥5000 | 84 | 20.6 | 267 | 21.3 |
| Health insurance | Yes | 199 | 48.8 | 631 | 50.3 |
| | No | 209 | 51.2 | 624 | 49.7 |
| Caregiver support | Yes | 319 | 78.2 | 991 | 78.9 |
| | No | 89 | 21.8 | 265 | 21.1 |

* Catholic, Jewish and Waaqeffannaa ** Somali, Afar, Gambela, and Diredewa *** Government

The factors that had an association with oral solid medication acceptance of older patients on bivariable analysis were educational status, number of oral solid medications, visual impairment, hypertension, diabetes mellitus, cardiac-related disorders, inflammation/allergy, infection/prophylaxis, medical conditions, frequency of visiting hospitals, general harm belief, necessity, concern beliefs, knowledge, size, shape, score line, packaging, texture, form, and labeling of medications. However, the number of oral solid medications, hypertension, infection/prophylaxis, frequency of visiting

**Table 2. Distributions of personal and therapy-related variables and corresponding oral solid medication consumption of older patients in Addis Ababa, Ethiopia.**

| Personal and therapy-related factors | Category | Frequency | Percent (%) | Respective OSM use (n) | Percent (%) |
|---|---|---|---|---|---|
| Alcohol consumption | Non-drinker | 303 | 74.5 | 938 | 74.7 |
| | Occasional drinker | 79 | 19.4 | 235 | 18.7 |
| | Daily drinker | 26 | 6.3 | 83 | 6.6 |
| Tobacco smoking | Non-smoker | 304 | 74.5 | 929 | 74.0 |
| | Ex-smoker | 75 | 18.4 | 231 | 18.4 |
| | Smoker | 29 | 7.1 | 96 | 7.6 |
| Chat chewing | Non-chewer | 228 | 55.9 | 701 | 55.8 |
| | Ex-chewer | 67 | 16.4 | 198 | 15.8 |
| | Chewer | 113 | 27.7 | 357 | 28.4 |
| Communication with healthcare providers | Good | 328 | 80.4 | 1008 | 80.3 |
| | Poor | 80 | 19.9 | 248 | 19.7 |
| Physical dependency | Independent | 199 | 48.8 | 600 | 47.8 |
| | Slightly dependent | 79 | 19.4 | 232 | 18.5 |
| | Moderately dependent | 47 | 11.5 | 140 | 11.1 |
| | Severely dependent | 33 | 8.1 | 117 | 9.3 |
| | Totally dependent | 50 | 12.3 | 167 | 13.3 |
| Dosing reminder | Not using any reminder | 216 | 52.9 | 626 | 49.8 |
| | Family/caregivers | 110 | 27.0 | 377 | 30.0 |
| | Alarms/phones/pillboxes | 67 | 16.4 | 205 | 16.3 |
| | Association with daily routines | 15 | 3.7 | 48 | 3.8 |
| Length of time on medication | ≤5 years | 209 | 51.2 | 566 | 45.1 |
| | 5–10 years | 116 | 28.4 | 375 | 29.9 |
| | 10–15 years | 48 | 11.8 | 183 | 14.6 |
| | >15 years | 35 | 8.6 | 132 | 10.5 |
| Number of medicines | 1 | 16 | 3.9 | 16 | 1.3 |
| | 2 | 120 | 29.4 | 240 | 19.1 |
| | 3 | 128 | 31.4 | 378 | 30.1 |
| | 4 | 96 | 23.5 | 383 | 30.5 |
| | ≥5 | 48 | 11.8 | 239 | 19.0 |

hospitals, general harm and necessity, personal belief, size, score line, texture, and labeling were significantly associated in multivariable analysis.

Compared to those taking only one oral solid medication, older patients taking 2, 3, 4, or 5 oral solid medications were 3.45, 7.16, 3.66, and 5.05 times more likely to accept oral solid medications, respectively (AOR = 3.45, 95% CI: 1.00–11.89, p = 0.050), 3 (AOR = 7.16, 95% CI: 1.99–25.78, p = 0.003), 4 (AOR = 3.66, 95% CI: 1.00–13.27, p = 0.048), and 5 (AOR = 5.05, 95% CI: 1.32–19.39, p = 0.018, respectively). Older patients with hypertension were 2.05 times more likely to have oral solid medication acceptance compared to those without hypertension (AOR = 2.05, 95% CI: 1.22–3.45, p = 0.007). In the opposite older patients taking anti-infective or prophylactic oral solid antibiotics were 57% less likely to accept oral solid medications compared to those without an infection (AOR = 0.43, 95% CI: 0.26–0.72, p = 0.001).

Older patients with a weak personal belief in the necessity of such medications were 49% less likely to accept them compared to those with a strong necessity belief (AOR = 0.49, 95% CI: 0.32–0.74, p < 0.001). Compared to oral solid medications ≤ 6 mm, those sized 6−9 mm and 9−12 mm were 6.5 and 2.6 times more likely to be accepted, respectively (AOR = 6.50, 95% CI: 3.84–10.99, p < 0.001, AOR = 2.6, 95% CI: 1.48–4.57, p < 0.001). Conversely,

**Table 3. Distributions of aging, clinical-related variables, and corresponding oral solid medication consumptions of older patients in Addis Ababa, Ethiopia.**

| Characteristics | | Frequency | Percent | Respective OSM use | Per-cent |
|---|---|---|---|---|---|
| Ageing related problem * | Visual impairment | 117 | 28.7 | 407 | 32.4 |
| | Swallowing difficulty | 113 | 27.7 | 403 | 32.1 |
| | Movement problem | 109 | 26.7 | 330 | 26.3 |
| | Memory problem | 65 | 15.9 | 220 | 17.5 |
| | Hearing problem | 24 | 6 | 104 | 8.3 |
| Disease conditions * | Hypertension | 176 | 43.1 | 631 | 50.2 |
| | Diabetes mallets | 96 | 23.5 | 360 | 28.7 |
| | Cardiac-related disorders (HF, IHD, stroke, hyperlipidemia, coagulation disorders) | 128 | 31.4 | 466 | 37.1 |
| | Asthma | 44 | 10.8 | 150 | 11.9 |
| | Pain, allergy, and inflammation | 92 | 22.5 | 287 | 22.9 |
| | CNS disorder (epilepsy, psychosis, depression, and convulsion) | 68 | 16.7 | 207 | 16.5 |
| | Endocrine disorder | 60 | 14.7 | 196 | 15.6 |
| | Chronic Liver Disease | 28 | 6.9 | 104 | 8.3 |
| | Infections | 52 | 12.7 | 182 | 14.5 |
| | Others (abdominal problem, kidney disease, and cancer | 46 | 11.3 | 170 | 13.5 |
| Severity of medical condition | Mild | 36 | 8.8 | 85 | 6.8 |
| | Moderate | 197 | 48.8 | 589 | 46.9 |
| | Severe | 175 | 42.9 | 582 | 46.3 |
| Hospital visits per year | Monthly | 227 | 55.6 | 727 | 57.9 |
| | Every two month | 26 | 6.4 | 80 | 6.4 |
| | Quarterly | 136 | 33.3 | 402 | 32 |
| | Bi annually | 19 | 4.7 | 47 | 3.7 |
| Satisfactions by HCP and health care services | Strongly unsatisfied | 8 | 2 | 40 | 3.2 |
| | Unsatisfied | 87 | 21.3 | 252 | 20.1 |
| | Neutral | 28 | 6.9 | 113 | 9 |
| | Satisfied | 265 | 64.9 | 811 | 64.6 |
| | Strongly satisfied | 20 | 4.9 | 48 | 3.2 |

*Multiple responses were possible.

medications ≥ 18 mm were 85% less likely to be accepted (AOR = 0.15, 95% CI: 0.06–0.35, p < 0.001). Regarding score lines, medications with a single score line were twice as likely to be accepted compared to those without (AOR = 1.99, 95% CI: 1.32–3.00, p = 0.001). Uncoated/rough oral solid medications were 73% less likely to be accepted than coated/smooth medications. (AOR = 0.27, 95% CI: 0.18–0.41, p < 0.001). Finally, medications with difficult labeling to read and understand were 72% less likely to be accepted than those with easy-to-understand labeling (AOR = 0.28, 95% CI: 0.10–0.79, p = 0.016). (Table 6)

## Discussion

This study investigated the impact of key physical characteristics of oral solid dosage forms (OSDFs) on medication acceptance among older patients, alongside sociodemographic, clinical, and behavioral correlates. By focusing on older populations in Ethiopia, a low- and middle-income country (LMIC), this research provides valuable context-specific

Table 4. Distributions of behavioral factors and corresponding oral solid medication consumptions of older patients in Addis Ababa, Ethiopia.

| Behavior | | | Frequency | Percentage | No of medications used | Percent | Mean value |
|---|---|---|---|---|---|---|---|
| Personal belief | General overuse belief | Strong | 148 | 36.3 | 526 | 41.9 | 2.6 (±0.8) |
| | | Weak | 260 | 63.7 | 730 | 58.1 | |
| | General harm belief | Strong | 204 | 50 | 666 | 53 | 1.8 (±0.7) |
| | | Weak | 204 | 50 | 590 | 47 | |
| | Necessity belief | Strong | 276 | 67.6 | 849 | 67.6 | 3.8 (±0.5) |
| | | Weak | 132 | 32.4 | 407 | 32.4 | |
| | Concern belief | Strong | 176 | 43.1 | 574 | 45.7 | |
| | | Weak | 232 | 56.9 | 682 | 54.3 | 2.3 (±0.6) |
| Religious belief | | Strong | 228 | 55.9 | 665 | 52.9 | 3.9 (±0.5) |
| | | Weak | 180 | 44.1 | 591 | 47.1 | |
| Cultural belief | | Strong | 88 | 21.6 | 292 | 23.2 | 2.7 (±0.9) |
| | | Fair | 88 | 21.6 | 300 | 23.9 | |
| | | Weak | 232 | 56.9 | 664 | 52.9 | |
| Knowledge | | Good | 251 | 61.5 | 731 | 58.2 | -- |
| | | Poor | 157 | 38.5 | 525 | 41.8 | |

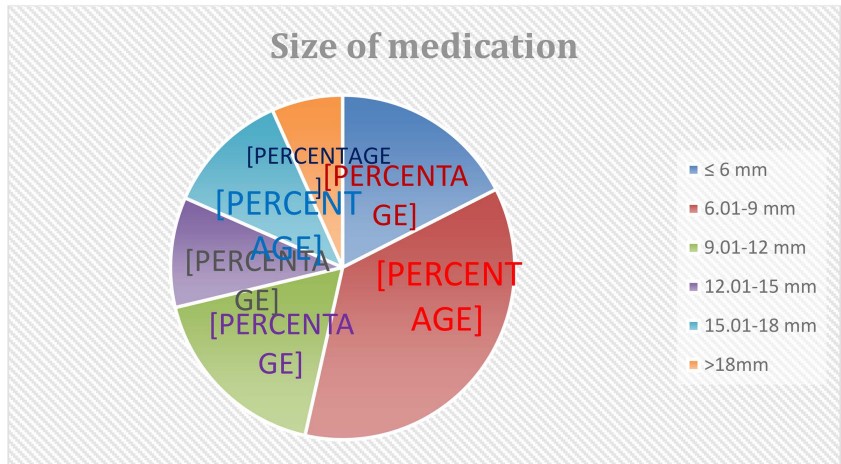

**Fig 2. Size distributions of oral solid medications of the study participants (older patients).** Proportions represent the share of all recorded OSM classified by maximum dimension (mm) into six bins (≤6, 6.01–9, 9.01–12, 12.01–15, 15.01–18, >18; n = 1256). Measurements were obtained from the summary of product characteristics (SmPC) and by measuring the size by using a measuring tool called a tablet and diameter measuring instrument" or a "tablet caliper with each unique product counted once per participant. Data were collected in Addis Ababa, Ethiopia, from May to July 2024 at five selected hospitals. Abbreviation: OSM, oral solid medication.

insights that extend existing literature, which has been largely derived from high-income settings. To our knowledge, this is among the first studies in sub-Saharan Africa to examine how medication design features influence acceptance in an aging population with limited formulation options and constrained healthcare infrastructure.

Overall, 75% of oral solid medications were accepted by older patients, a level comparable to findings from Germany (80%) [5]. However, the determinants of acceptance in our setting highlight context-specific barriers and opportunities. The study revealed that oral solid formulations measuring between 6 and 12 millimeters were most well-accepted by the older participants. In contrast, those 18 millimeters or larger were significantly less accepted compared with ≤6 mm.

**Table 5. Characteristics of oral solid medications used by older patients in Addis Ababa, Ethiopia.**

| Characteristics of OSM: | | Frequency | Percentage |
|---|---|---|---|
| Shape | Round | 920 | 73.2 |
| | Oval | 217 | 17.3 |
| | Oblong | 119 | 9.5 |
| Color | White/off white | 754 | 60 |
| | Yellow | 166 | 13.2 |
| | Two colors | 97 | 7.72 |
| | Orange/green | 46 | 3.7 |
| | Peach/pink/red | 140 | 11.2 |
| | Blue brown | 53 | 4.2 |
| Taste | Bitter taste | 730 | 58.1 |
| | Sweet | 167 | 13.3 |
| | Neutral | 268 | 21.4 |
| | Salty/sour/metallic | 91 | 7.2 |
| Score line | None | 439 | 35 |
| | Single | 749 | 59.6 |
| | Partial | 56 | 4.5 |
| | Multiple | 12 | 1 |
| Texture | Coated/smooth | 466 | 37.1 |
| | Uncoated/rough | 768 | 61.1 |
| | Chewable | 8 | 0.6 |
| | Disintegrating | 12 | 1 |
| Smell | Pleasant | 352 | 28 |
| | Neutral | 355 | 26.7 |
| | Unpleasant | 569 | 45.3 |
| Form | Tablet | 1154 | 91.9 |
| | Capsule | 102 | 8.1 |
| Labeling | Not understanding the language of labeling for older patients | 1008 | 80.3 |
| | Difficult to read and understand for older patients | 168 | 13.4 |
| | Ease to read and understand for older patients | 80 | 6.4 |

This stark difference in acceptance rates between the smaller (6–12 mm) and larger (≥18 mm) oral solid medications underscores the importance of carefully considering medication size when developing products intended for the older population. As people age, they often experience diminished dexterity, weakened swallowing function, and other physiological changes that can make larger, bulkier oral solid drugs challenging to manage [10].

The results of this study are consistent with findings from previous research conducted in other countries and a local study. A study in the UK found that tablets greater than 11 mm and capsules larger than 13 mm had acceptability issues due to swallowing difficulties, especially among older patients with existing swallowing problems [19]. Similarly, studies in France and the United Kingdom have reported that 83% of patients failed to accept oral solid medications greater than 10 mm in size [20]. Another study conducted in Japan found that extremely large tablets (>22 mm) were negatively accepted by older patients [21]. A local study in Gondar town also revealed a preference for small to medium-sized oral solid formulations over larger tablets and capsules [22].

The available evidence suggests that medications with small to medium-sized oral solid medications tend to have higher acceptance rates among older patients compared to larger-sized tablets and capsules. This underscores the

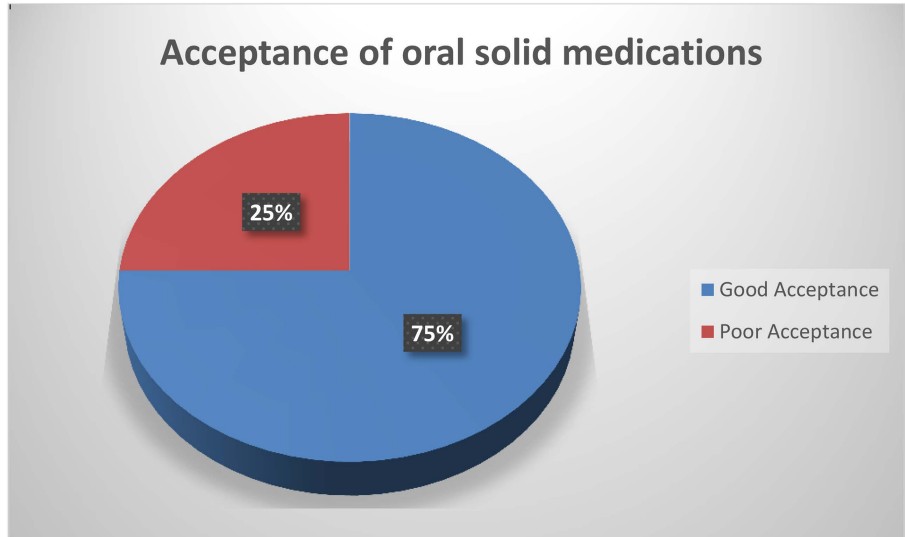

**Fig 3. Acceptance of oral solid medications among older participants.** Pie slices show the proportion of medications classified as accepted versus not accepted based on patient ratings of each medication (n = 1,256 medications). Acceptance was assessed using a 5-point Likert scale for each medication; ratings of 3 (neutral) or higher were classified as accepted, and ratings below 3 as not accepted. Data were collected in Addis Ababa, Ethiopia, using questionnaire/structured interview during from May to July 2024 in five selected hospitals. Abbreviation: OSM, oral solid medication.

importance of optimizing medication size to meet the unique physiological needs and capabilities of the geriatric population, thereby improving adherence and overall treatment outcomes. In our context, where alternative formulations such as smaller tablets, dispersible, chewable, or liquid forms are often unavailable or unaffordable, the implications are particularly significant. Physicians and pharmacists frequently have limited choices when prescribing, emphasizing the need for improved formulation diversity in low and middle income countries (LMIC) markets.

This study also highlighted the importance of score lines on oral solid medications, particularly tablets. The findings revealed that 78.4% of medications with single score lines were significantly accepted by older patients compared to tablets without score lines.

This suggests that the presence of score lines on tablets plays an important role in patient acceptance, particularly for older patients who may need to divide or break their medication into smaller or more manageable portions. Tablets with score lines allow patients to confidently divide the tablet into equal parts, which can be crucial for ensuring proper dosing and medication adherence [11].

This finding is even more substantial than the results of a previous study conducted in the Netherlands, which found that only 64% of older patients accepted tablets with score lines [23]. The higher acceptance rate (78.4%) observed in the current study further emphasizes the importance of incorporating score lines into the design of oral solid formulations intended for the geriatric population. So, by including score lines on tablets, pharmaceutical manufacturers can enhance the ease of use and overall acceptability of their products among older patients. This design feature can contribute to improved patient satisfaction, medication adherence, and ultimately, better therapeutic outcomes for this vulnerable population. Score lines facilitate dose adjustment and ease of swallowing, which is particularly important in older patients with dysphagia. However, the availability of scored tablets remains limited in Ethiopia, underscoring a gap between patient needs and market availability. Incorporating this feature more broadly could improve usability without major manufacturing cost implications.

Other significant oral solid medication physical characteristics which significantly associated with acceptance in this study was texture. 80.3% of coated tablets were significantly accepted by older patients. Older patients tend to correlate

**Table 6. Factors associated with oral solid medication acceptance among older patients in Addis Ababa, Ethiopia, 2025.**

| Variable | | Acceptance of medications | | COR | P-value (95% CI) | AOR | P-value (95% CI) |
|---|---|---|---|---|---|---|---|
| | | Yes | No | | | | |
| Level of education | Illiterate | 494 | 203 | 0.66(0.44,0.99) | 0.042 | 1.18(0.61,2.25 | 0.627 |
| | Primary school | 173 | 29 | 1.6(0.932.76) | 0.087 | 2.01(0.97,4.51) | 0.059 |
| | Secondary school | 148 | 44 | 0.91(0.55,1.50) | 0.70 | 1.40(0.70,2.78) | 0.346 |
| | College and above | 130 | 35 | 1 | 1 | 1 | 1 |
| Number of oral solid medications | 1 | 8 | 8 | 1 | 1 | 1 | 1 |
| | 2 | 180 | 60 | 3.00(1.08-8.34) | 0.035 | 3.45(1.00,11.89) | **0.050** |
| | 3 | 295 | 83 | 3.56(1.30-9.80) | 0.014 | 7.16(1.99,25.78) | **0.003** |
| | 4 | 279 | 104 | 2.70(0.98-7.33) | 0.054 | 3.66(1.00,13.27) | **0.048** |
| | 5 | 183 | 56 | 3.27(1.17-9.11) | 0.024 | 5.05(1.32,19.39) | **0.018** |
| Visual impairment | Yes | 323 | 84 | 0.71(0.54-0.96) | 0.019 | 1.22(0.78,1.91) | 0.39 |
| | No | 622 | 227 | 1 | 1 | 1 | 1 |
| Hypertension | Yes | 512 | 119 | 1.91(1.47,2.48) | <0.001 | 2.05(1.22,3.45) | **0.007** |
| | No | 433 | 192 | 1 | 1 | 1 | 1 |
| DM | Yes | 288 | 72 | 1.46(1.08,1.96) | 0.014 | 0.99(0.57,1.77) | 0.99 |
| | No | 657 | 239 | 1 | 1 | 1 | 1 |
| Cardiac related disorders | Yes | 574 | 95 | 1.47(1.11,1.93) | 0.006 | 0.72(0.44,1.20) | 0.20 |
| | No | 371 | 216 | 1 | 1 | 1 | 1 |
| Allergy | Yes | 199 | 88 | 0.68(0.51-0.91) | 0.009 | 1.12(0.68,184) | 0.65 |
| | No | 746 | 223 | 1 | 1 | 1 | 1 |
| Infection | Yes | 112 | 70 | 0.44(0.32,0.62) | <0.001 | 0.43(0.26,0.72) | **0.001** |
| | No | 833 | 241 | 1 | 1 | 1 | 1 |
| Medical condition | Mild | 63 | 22 | 1.1(0.66,1.85) | 0.707 | 0.82(0.40,1.67) | 0.593 |
| | Moderate | 462 | 127 | 1.40(1.07,1.83) | 0.013 | 0.83(0.57,0.21) | 0.336 |
| | Severe | 420 | 162 | 1 | 1 | 1 | 1 |
| Frequency of visiting hospitals | Monthly | 526 | 201 | 1 | 1 | 1 | 1 |
| | Every two months | 50 | 30 | 0.64(0.39,1.03) | 0.066 | 1.39(0.70,2.78) | 0.348 |
| | Quarterly | 340 | 62 | 2.1(1.53,2.88) | <0.001 | 2.00(1.27,3.17) | **0.003** |
| | Biannually | 29 | 18 | 0.62(0.33,1.13) | 0.119 | 2.26(0.91,5.61) | 0.079 |
| General harm belief | Strong believer | 479 | 187 | 0.68(0.53-0.88) | 0.004 | 0.65(0.41,1.02) | 0.062 |
| | Weak believer | 466 | 124 | 1 | 1 | 1 | 1 |
| Necessity belief | Strong believer | 682 | 167 | 1 | 1 | 1 | 1 |
| | Weak believers | 263 | 144 | 0.45(0.34,0.58) | <0.001 | 0.49(0.32,0.74) | **<0.001** |
| Concern belief | Strong believer | 403 | 171 | 0.61(0.47-0.79) | <0.001 | 0.99(0.66,1.48) | 0.952 |
| | Weak believer | 542 | 140 | 1 | 1 | 1 | 1 |
| Knowledge belief | Good knowledge | 578 | 153 | 1 | 1 | 1 | 1 |
| | Poor knowledge | 367 | 158 | 0.62(0.48,0.80) | | 0.64(0.40,1.04) | 0.070 |
| size | ≤ 6 mm | 136 | 84 | 1 | 1 | 1 | 1 |
| | 6-9 mm | 416 | 36 | 7.14(4.62,11.04) | <0.001 | 6.50(3.84,10.99) | **<0.001** |
| | 9-12 mm | 188 | 35 | 3.32(2.11,5.21) | <0.001 | 2.6(1.48,4.57) | **<0.001** |
| | 12-15 mm | 82 | 48 | 1.06(0.67,1.65) | 0.814 | 0.63(0.33,1.22) | 0.172 |
| | 15-18 mm | 95 | 52 | 1.13(0.73,1.74) | 0.585 | 0.56(0.24,1.29) | 0.173 |
| | ≥ 18 mm | 28 | 56 | 0.31(0.18,0.52) | <0.001 | 0.15(0.06,0.35) | **<0.001** |

*(Continued)*

| Variable | | Acceptance of medications | | COR | P-value (95% CI) | AOR | P-value (95% CI) |
|---|---|---|---|---|---|---|---|
| | | Yes | No | | | | |
| Shape | Round | 741 | 179 | 1 | 1 | 1 | 1 |
| | Oval | 129 | 88 | 0.35(0.23,0.49) | <0.001 | 1.17(0.64,2.12) | 0.613 |
| | Oblong | 75 | 44 | 0.41(0.27,0.62) | <0.001 | 1.97(0.64, 6.03) | 0.236 |
| Score line | None | 298 | 141 | 1 | 1 | 1 | 1 |
| | Single | 587 | 162 | 1.71(1.32,2.24) | <0.001 | 1.99(1.32,3.00) | **0.001** |
| | Partial/multiple | 60 | 8 | 2.84(1.31,6.20) | 0.008 | 2.29(0.80,6.62) | 0.124 |
| Packaging | Blister | 648 | 247 | 1 | 1 | 1 | 1 |
| | Bottle | 233 | 60 | 1.48(1.08,2.04) | 0.016 | 0.73(0.47,1.12) | 0.182 |
| | Plastic bag | 64 | 4 | 3.81(1.35,10.8) | 0.012 | 0.53(0.17,1.660 | 0.273 |
| Texture | Coated/smooth | 375 | 92 | 1 | 1 | 1 | 1 |
| | Uncoated/rough | 554 | 215 | 0.63(0.48,0.83) | 0.001 | 0.27(0.18,0.41) | **<0.001** |
| | Chewable/disintegrating | 16 | 4 | 0.98(0.32,3.01) | 0.974 | 0.44(0.12,1.61) | 0.214 |
| Form | Tablet | 878 | 276 | 1 | 1 | 1 | 1 |
| | Capsule | 67 | 35 | 0.60(0.39,0.93) | 0.021 | 0.94(0.32,2.75) | 0.915 |
| Labeling | Not understanding the language of labeling for older patients | 751 | 257 | 0.28(0.13-0.62) | 0.002 | 0.54(0.19,1.53) | 0.245 |
| | Difficult to read and understand for older patients | 121 | 47 | 0.25(0.11-0.58) | 0.001 | 0.28(0.10,0.79) | **0.016** |
| | Ease to read and understand for older patients | 73 | 7 | 1 | 1 | 1 | 1 |

COR = crude odds ratio; AOR = adjusted odds ratio; 95% CI = 95% confidence interval. Bivariable logistic regression was used to calculate CORs, and multivariable logistic regression with cluster-robust standard errors was used to calculate AORs. *p < 0.05 indicates statistical significance.

coated/smooth medications with lower side effects, higher efficacy, and higher quality, even if some of these perceptions are not always supported by empirical evidence.

This finding aligns with a study conducted in the United Kingdom, which found that 79% of older patients preferred film-coated tablets over uncoated acyclovir tablets, citing the "smoother" and "easier to swallow" nature of the coated formulation [23]. Similarly, a study in the Netherlands revealed that 81.5% of older patients accepted coated oral solid medications, reporting that the smooth/coated formulations helped to solve swallowing-related problems and prevented the medication from getting stuck in the throat [11]. Other research from various parts of the world has also confirmed the higher acceptance of coated tablets or capsules compared to uncoated formulations [24,25]. Coating improves swallow ability and patient perception of quality, as also observed in studies from the UK and the Netherlands [32,33]. In Ethiopia, most locally manufactured tablets remain uncoated due to constraints in coating technology, raw material access, and production costs. Manufacturers should balance patient-centered design with technical feasibility, considering factors such as dose load, compression strength, coating material availability, and production scalability.

Manufacturing feasibility considerations are critical when interpreting these findings. Larger tablets may be necessary to deliver high-dose medications, constraining size reduction without compromising efficacy. Incorporating coatings requires specialized machinery, raw materials, and quality control, which can be challenging in LMICs [26]. Tablets must also maintain sufficient compression strength to prevent breakage, and factors such as dose load, excipient compatibility, and stability can limit implementation of patient-preferred characteristics [27]. Therefore, while small, coated, and scored tablets are preferred by patients, pharmaceutical developers must balance patient preferences with technical feasibility,

cost, and production constraints. Strategies such as prioritizing widely used medications for patient-centered design or exploring multi-particulate and film-coated high-dose systems may help reconcile these constraints.

Another factor significantly related to oral solid medication acceptance was package labeling. Medication package labeling which is difficult to read and understand significantly lowers medication acceptance by 72%. This means oral solid medications reported as easy to read and understand were more acceptable by patients. This finding is slightly higher than a study conducted in the Netherlands where 63% of older patients greater than 70 reported problems with reading and understanding the medication's packaging instructions for use. This disparity could be due to language differences. In Ethiopia, English is not the predominant native language, yet medication packaging and instructions are often printed in English. For older Ethiopian patients, the use of a non-local language on product labeling can present a significant obstacle to comprehending the proper usage of their medications. Older patients may struggle to read and understand the instructions if they are not presented in their native tongue. Labeling of packaging in the local language and bold/underlining the most important information may increase the acceptance of medications [11]. Using clear, locally translated, and age-friendly labeling could improve medication understanding and adherence.

This study also examined other physical characteristics of oral solid medications, such as shape, color, packaging, form, smell, and taste. However, these factors were not found to be significantly associated with patient acceptance. While some trends were observed, such as a preference for white-colored, round-shaped, and tablet-form medications among older patients, these findings were not statistically significant. Therefore, the researchers did not delve into these characteristics in detail, as the focus of this study was on the more significant factors that influenced medication acceptance and adherence.

In terms of dosage form distribution, tablets were the predominant form, followed by capsules, which included both hard and soft gelatin types. No patients were taking powder-only oral solid dosage forms, as powders require reconstitution and do not have defined size, shape, or score lines like tablets and capsules. This focus on tablets and capsules allowed for the assessment of key physical characteristics; such as size, shape, coating, and score lines and others that influence patient acceptance. While soft gelatin capsules are generally easier to swallow than some tablets, they were less commonly used in this setting, reflecting prescribing practices and availability rather than patient preference.

This study also examined the influence of sociodemographic, clinical, and behavioral factors on the acceptance and adherence of oral solid medications among older patients. One key finding was that as the number of oral solid medications taken by older patients increased, their acceptance of these medications also significantly increased. Usually, older patients taking multiple medications frequently have multiple medical conditions. Therefore, older patients with multiple chronic medical conditions, and who are therefore taking a higher number of medications, often require a combination of medications to manage their symptoms effectively and those groups of patients may be highly motivated to improve their quality of life and manage their symptoms. Another possible reason could be older patients with polypharmacy often have more frequent interactions with healthcare providers due to the complexity of their medical conditions or taking multiple medications can become a routine or habit for older patients [28].

This study also found that the acceptance of oral solid medications was influenced by the type of medication being taken by older patients. Specifically, this finding revealed that older patients taking antihypertensive medications had nearly two times greater acceptance of oral solid medications compared to those who were not taking antihypertensive drugs. This could be attributed to the fact that many antihypertensive medications are smaller to medium-sized, round-shaped formulations, which may be more easily accepted by older patients [29].

Additionally, the study found that older patients taking anti-infective or prophylactic medications had a 56% lower acceptance of oral solid medications compared to those not taking such medications. This could be due to the larger size, oval or oblong shape, and potentially bitter or unpleasant taste often associated with anti-infective medications, which may present challenges for older patients in terms of swallowing and overall acceptance [30].

                                                              

This study also found that older patients who received quarterly follow-up care had higher acceptance of their medications compared to those who received monthly follow-up care. This could be due to the fact that older patients who only need quarterly follow-up visits may be in a more stable health condition, and thus, more willing and able to properly take their medications as prescribed. The less frequent visits to the healthcare facility may also be associated with a higher quality of life, which could positively influence their acceptance and adherence to their medication regimens [31].

This study also found that medication acceptance was significantly influenced by some behavioral factors. Patients who strongly believed in the necessity of their medications had 55% better acceptance of the oral solid formulations compared to those who did not perceive a strong need for the medication. This finding aligned with previous studies suggesting that people who do not see a need for a medication or have reservations about taking it are more likely to forget or become less adherent and have poor acceptance.

In summary, this study contributes to the growing evidence that medication design features critically affect usability and acceptance, particularly among older adults in resource-limited settings. It highlights the urgent need for context-appropriate, age-friendly formulations that consider both patient preferences and manufacturing feasibility. Pharmaceutical industries in LMICs should be encouraged to adopt coating technologies, optimize tablet size, and improve labeling, while health systems should promote prescriber awareness and encourage patient involvement in design feedback. Future research should also explore alternative formulations, including dispersible tablets and soft gels, to expand therapeutic accessibility for older populations.

## Limitations of the study

This study faced some limitations, particularly since it is the first of its kind in Ethiopia. The first limitation pertains to the study area; it would have been beneficial to include at least three regions to enhance the generalizability of the findings to the broader Ethiopian population. The second limitation is this research did not supported by experimental procedures. Unfortunately, these limitations could not be addressed due to budget constraints.

## Conclusion

While no single medication can meet all the needs of older patients, this study found that certain characteristics can enhance the overall acceptance to oral solid medications. Older patients tend to prefer oral solid medications that are small to medium-sized, specifically those ranging from 6 to 9 mm in diameter, and round in shape. Extremely small medications (less than 6 mm) or large (greater than 18 mm) are better to be avoided for this group of patients. Additionally, oral solid medications with a single score line and those that are coated or have a smooth texture are associated with better acceptance. White-colored medications also tend to be favored over others.

## Supporting information

**S1 File. Oral solid medication acceptance among older patients in Addis Ababa, Ethiopia (Questionnaire).**
(DOCX)

**S2 File. Minimal anonymized dataset underlying the findings.**
(XLSX)

**S3 File. Variable definitions and value labels.**
(PDF)

## Acknowledgments

The authors want to extend their acknowledgement to Jimma University, Institute of Health, Faculty of Health Sciences, School of Pharmacy, for allowing to conduct this research. Additionally, the authors would like to thank Addis Ababa City Administration Health Bureau, Tikur Anbessa Specialized Hospital, St. Paul's Hospital Millennium Medical College,

 

Zewuditu Memorial Hospital, Menelik II Referral Hospital, and Yekatit 12 Hospital, as well as all study participants and data collectors, for their support and consent.

## Author contributions

**Conceptualization:** Mohammed Chane Assefa, Getahun Paulos, Dereje Kebebe Borga.

**Data curation:** Mohammed Chane Assefa, Getahun Paulos, Dereje Kebebe Borga.

**Formal analysis:** Mohammed Chane Assefa.

**Investigation:** Mohammed Chane Assefa, Getahun Paulos.

**Methodology:** Mohammed Chane Assefa, Dereje Kebebe Borga.

**Project administration:** Fikadu Ejeta, Dereje Kebebe Borga.

**Resources:** Fikadu Ejeta.

**Software:** Mohammed Chane Assefa.

**Supervision:** Fikadu Ejeta, Dereje Kebebe Borga.

**Validation:** Fikadu Ejeta, Dereje Kebebe Borga.

**Visualization:** Dereje Kebebe Borga.

**Writing – original draft:** Mohammed Chane Assefa.

**Writing – review & editing:** Mohammed Chane Assefa.

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
