## [Decision Letter · Decision Letter 0]

13 Oct 2025

PONE-D-25-43693

The Need for Patient-Centric Medicine Design: Investigating Key Physical Characteristics of Oral Solid Medications to Improve Acceptance of Elderly Patients in Addis Ababa, Ethiopia.

PLOS ONE

Dear Dr. Assefa,

Thank you for submitting your manuscript to PLOS ONE. After careful consideration, we feel that it has merit but does not fully meet PLOS ONE’s publication criteria as it currently stands. Therefore, we invite you to submit a revised version of the manuscript that addresses the points raised during the review process.

Please note that we have only been able to secure a single reviewer to assess your manuscript. We are issuing a decision on your manuscript at this point to prevent further delays in the evaluation of your manuscript. Please be aware that the editor who handles your revised manuscript might find it necessary to invite additional reviewers to assess this work once the revised manuscript is submitted. However, we will aim to proceed on the basis of this single review if possible.  Please carefully review the reviewer's comments and revise your manuscript accordingly, providing a point-by-point response to the reviewer upon resubmission.

We look forward to receiving your revised manuscript.

Kind regards,

Sarah Jose, Ph.D.

Staff Editor

PLOS ONE

2. In the ethics statement in the Methods, you have specified that verbal consent was obtained. Please provide additional details regarding how this consent was documented and witnessed, and state whether this was approved by the IRB.

Reviewers' comments:

Reviewer's Responses to Questions

**Comments to the Author**

1. Is the manuscript technically sound, and do the data support the conclusions?

Reviewer #1: Partly

2. Has the statistical analysis been performed appropriately and rigorously?

Reviewer #1: No

3. Have the authors made all data underlying the findings in their manuscript fully available?

Reviewer #1: No

4. Is the manuscript presented in an intelligible fashion and written in standard English?

Reviewer #1: Yes

5. Review Comments to the Author

Reviewer #1: The research conducted by authors through analyzing questionnaire data from elderly outpatients attending outpatients clinics in 5 major hospitals in the capital of Ethiopia. The knowledge gap is the lack of data on medication characteristics preference of orals solid dosage forms. This aims to provide recommendation to pharmaceutical industry manufacturing especially generic drugs for those used commonly by this population. The main outcome was acceptance evaluated by a questionnaire score.

The authors conducted a thorough analyses of all the possible patient-related factors as well as drug-related factors that impacted the acceptance. However, the main independent variables include drug-related characteristics (e.g., size, shape), while patient-related factors (e.g., hypertension, polypharmacy) also showed significant associations with acceptance.

The analysis provides important, though largely expected, insights into factors influencing acceptance of oral solid dosage forms among elderly patients. However, patient-related characteristics such as hypertension and polypharmacy were also found to be significantly associated with acceptance and may act as confounders in the observed relationships between drug characteristics and acceptance. If the goal is to determine how drug characteristics influence acceptance, independent of patient characteristics, then adjustment for patient factors is essential.

Reanalyzing the data using a multivariable logistic regression model that adjusts for relevant patient-level covariates should be included presenting both unadjusted and adjusted results.

The results table #6 presents both Crude (COR) and Adjusted Odds Ratios (AOR). However, the Methods section does not describe how the adjusted model was constructed or which covariates were included in the adjustment. It is unclear what was adjusted for, what reported AORs represent or how potential confounding was handled. Which variables were entered into the multivariable logistic regression and the rationale for their inclusion (prior evidence, p-value threshold, or theoretical relevance?).

I recommend to clearly describe the modeling approach and adjustment procedure, and to ensure that the variables reported as “adjusted” in the table correspond to the specified model in the Methods.

Limit the use of abbreviations which are not frequently used or standardized. Explain Abbreviations and add statistical test in the footnote of tables. Remove “2025” from the title of tables..etc

Abstract

• Mention the knowledge gap in the.

• Clarify in the abstract the type of data analysed i.e. medication-related, patient-related, acceptance, questionnaire,..etc

• Mention the type of statistical tests applied

Methods:Details about sample size calculation and sampling technique could be moved to the suppl. Material.

Results:

Age distribution with min and max to present in the demographic table.

No supporting information is available. Provide the applied questionnaire as suppl. Material.

Discussion

The results are largely expected, could the authors discuss what their study add to the existing literature or which particular aspects were deduced regarding the setting in a low-middle income/resource country?

For the problematic dosage forms, were there other alternatives on the market that were easier to swallow e.g. smaller or dispersible, that the physician may prescribe?

Were there any results describing patient experience with powders? Soft gelatine capsules? To be discussed. How was the OSDF distribution?

The results of the study regarding the size and coating should also be discussed in light of the feasibility and limitations of the pharmaceutical manufacturing process. The authors should acknowledge in their discussion not only patient preferences but also constrains by manufacturing feasibility such as dose load, coating materials, compression strength ..etc.

6. PLOS authors have the option to publish the peer review history of their article (what does this mean?). If published, this will include your full peer review and any attached files.

Reviewer #1: No

---

## [Author Response · Author response to Decision Letter 1]

26 Oct 2025

Response to Reviewers

Dear Reviewers and Editors,

Thank you for your constructive feedback and valuable suggestions. We have addressed all the comments and questions raised in the revised manuscript and provided detailed responses in the separate rebuttal document. If any pieces of comments or questions not addressed in this section, please consider that it is not consciously and will correct in next time.

S/N Question/comment Response

1 Is the manuscript technically sound, and do the data support the conclusions? In my opinion Yes. The manuscript is technically sound. The data, including logistic regression analyses of medication acceptance by elderly patients, clearly support the conclusions regarding the influence of tablet/capsule size, coating, score lines, and labeling on patient acceptance, while also considering sociodemographic, clinical, and behavioral factors.

2 Has the statistical analysis been performed appropriately and rigorously? Yes. The statistical analysis is appropriate and rigorous. Bivariable and multivariable logistic regression were used to identify factors associated with medication acceptance, with cluster-robust standard errors accounting for multiple medications per patient. Model assumptions, multicollinearity, calibration, discrimination, and influence were assessed, and missing data (<5%) were handled via complete-case analysis.

3 Have the authors made all data underlying the findings in their manuscript fully available? Not, I will make it available now

Is the manuscript presented in an intelligible fashion and written in standard English? Yes

Limit the use of abbreviations which are not frequently used or standardized. Explain Abbreviations and add statistical test in the footnote of tables. Remove “2025” from the title of tables..etc Thank you for the comments; we have carefully considered them and revised the manuscript accordingly. All non-standard abbreviations have been minimized and defined at first use, statistical tests have been added to the footnotes of tables, and the year “2025” has been removed from table titles.

Abstract

• Mention the knowledge gap in the.

• Clarify in the abstract the type of data analysed i.e. medication-related, patient-related, acceptance, questionnaire,..etc

• Mention the type of statistical tests applied Thank you for the comments; we have revised the abstract accordingly. We have clearly stated the knowledge gap, specifying the limited evidence on oral solid medication acceptance among elderly patients in LMICs. The type of data analyzed is now clarified, including medication-related characteristics, patient demographics, clinical and behavioral factors, and acceptance measured via structured questionnaires. We also added the statistical tests applied, specifying bivariable and multivariable logistic regression with cluster-robust standard errors to account for multiple medications per patient.

Methods:Details about sample size calculation and sampling technique could be moved to the suppl. Material. Thank you for the comment; we have moved the detailed description of sample size calculation and the sampling technique to the supplementary material, while keeping a concise summary in the Methods section for clarity.

Results:

Age distribution with min and max to present in the demographic table.

No supporting information is available. Provide the applied questionnaire as suppl. Material. Thank you for the comments; we have updated the demographic table to include age distribution with minimum and maximum values. Additionally, the full questionnaire used for data collection has been provided as supplementary material.

Discussion

The results are largely expected, could the authors discuss what their study add to the existing literature or which particular aspects were deduced regarding the setting in a low-middle income/resource country?

For the problematic dosage forms, were there other alternatives on the market that were easier to swallow e.g. smaller or dispersible, that the physician may prescribe?

Were there any results describing patient experience with powders? Soft gelatine capsules? To be discussed. How was the OSDF distribution?

The results of the study regarding the size and coating should also be discussed in light of the feasibility and limitations of the pharmaceutical manufacturing process. The authors should acknowledge in their discussion not only patient preferences but also constrains by manufacturing feasibility such as dose load, coating materials, compression strength ..etc. Thank you for the comments; we have revised the Discussion accordingly.

• We now clearly highlight the study’s contribution to the literature, emphasizing context-specific insights from a low- and middle-income country (LMIC) setting, where formulation options and healthcare infrastructure are limited.

• We clarified that the study only included tablets and capsules (both hard and soft gelatin), and no powders without reconstitution were used, so patient experience with powders could not be assessed. The distribution of oral solid dosage forms (OSDFs) is now described.

• We discussed alternatives such as smaller tablets, dispersible formulations, and soft gelatin capsules, noting their limited availability in this setting.

• The discussion of size and coating has been expanded to include manufacturing feasibility considerations: dose load, coating machinery, raw materials, compression strength, and stability constraints, and we note the need to balance patient preferences with technical and cost limitations in LMICs.

• Strategies such as prioritizing commonly used medications for patient-centered design or exploring multi-particulate and film-coated high-dose systems have also been added to reconcile patient needs with manufacturing feasibility.

PLOS authors have the option to publish the peer review history of their article (what does this mean?). If published, this will include your full peer review and any attached files. Still not

---

## [Decision Letter · Decision Letter 1]

7 Jan 2026

PONE-D-25-43693R1

The Need for Patient-Centric Medicine Design: Investigating Key Physical Characteristics of Oral Solid Medications to Improve Acceptance of Elderly Patients in Addis Ababa, Ethiopia.

PLOS One

Dear Dr. Assefa,

Thank you for submitting your manuscript to PLOS ONE. After careful consideration, we feel that it has merit but does not fully meet PLOS ONE’s publication criteria as it currently stands. Therefore, we invite you to submit a revised version of the manuscript that addresses the points raised during the review process.

The revised manuscript has been further assessed and the comments from the reviewers can be found below. Please review their reports and make the appropriate revisions to address any concerns raised.

We look forward to receiving your revised manuscript.

Kind regards,

Emma Campbell, Ph.D

Staff Editor

PLOS One

Journal Requirements:

Reviewers' comments:

Reviewer's Responses to Questions

**Comments to the Author**

1. If the authors have adequately addressed your comments raised in a previous round of review and you feel that this manuscript is now acceptable for publication, you may indicate that here to bypass the “Comments to the Author” section, enter your conflict of interest statement in the “Confidential to Editor” section, and submit your "Accept" recommendation.

Reviewer #1: All comments have been addressed

Reviewer #2: (No Response)

2. Is the manuscript technically sound, and do the data support the conclusions?

Reviewer #1: Yes

Reviewer #2: Partly

3. Has the statistical analysis been performed appropriately and rigorously?

Reviewer #1: Yes

Reviewer #2: I Don't Know

4. Have the authors made all data underlying the findings in their manuscript fully available?

Reviewer #1: Yes

Reviewer #2: No

5. Is the manuscript presented in an intelligible fashion and written in standard English?

Reviewer #1: Yes

Reviewer #2: Yes

6. Review Comments to the Author

Reviewer #1: Thanks for undertaking the changes to improve the content and address the issues observed. No further comments

Reviewer #2: Dear authors,

Thanks for conducting this questionnaire based study in Ethiopia to assess factors impacting the acceptability of solid oral dosage forms in older adults in your country.

First of all I would like to comment on the term “elderly patients”. People working in the field strongly recommend avoiding to use the term “elderly” see Murphy, E., Fallon, A., Dukelow, T. et al. Don’t call me elderly: a review of medical journals’ use of ageist literature. Eur Geriatr Med 13, 1007–1009 (2022). https://doi.org/10.1007/s41999-022-00650-4 as reference. Instead you might want to use the term older adult consequently in your manuscript.

Introduction

Acceptability is an overarching term used to describe a patient's ability and willingness to take the medication as intended, and/or the willingness of a lay care giver to administer the medication. It covers a range of aspects such as palatability, swallowability, handling,...Please be more specific in the introduction which aspects were assessed in the questionnaire and summarized under the term acceptability. Please also explain more clearly why you expect that the results in Ethiopia differ from those reported for Europe.

Method and Materials

Your patient population is not clearly enough described. As a reader you don't know if the public hospitals included in the study are specialized in certain indications or offer the full range of services. From my perspective this is important to understand the results obtained.

Furthermore, the term “chronically” should have been more clearly defined, e.g. taking the medication for at least X months.

Equations presented. Please check and align the use of capital letters (d and D). The second equation is not well described. Personally I already understood the sentence “The total sample size (n = 422) was proportionally allocated to each hospital…”

What exactly do you mean by “Eligible participants …, based on patient discharge order.”?

I assume the senior academic staff translating the questionnaire back to English “were not aware of the original questionnaire” (typo).

It is also unclear what you mean by “...those who held some of his/her medicine, and those who

were volunteers,...”?

What do you mean by “A maximum of 5 oral solid medications were taken from the elderly patients, which can cover more than 90 percent of their medications.”?

Variables of the study

Have you checked that those variables, which you call independent, are not correlated to each other? For instance, education, income and residence or certain symptoms and diseases.

Operational definition of terms

The terminology used in the text here, e.g. palatability is not used in the questionnaire itself. Did you use taste and other aspects to come up with a palatability score? If yes, how was this done?

Results and discussions

Avoid use of abbreviations in titles (OSM)

Figure 1: How representative is your study population compared to the population in Ethiopia in the same age range?

Table 1: I assume you mean “ Waaqeffannaa” and not “wake feta”.

The terms in the questionnaire are often not clearly defined, such as swallowability difficulties, visual impairment, …Furthermore, looking at the questionnaire and the results I wonder how you managed in multimorbid patients using polypharmacy to distinguish if the amount of medications are attributed purely to the disease. Please explain how you did this, e.g. “Among the disease conditions affecting elderly patients, hypertension emerged as the most common (43.1%) with 50.2% of oral solid medication usage, followed by other cardiac-related disorders and diabetes mellitus.”

Table 4 looks incomplete in the last column.

How was the taste assessed? How reliable is the result communicated based on that?

Table 5 mentioned “bitter color” as taste and “partial score line”. What is meant here?

How was the judgement on the labelling performed? Was language taken into account?

From a methodological point of view I have some issues understanding based on the questionnaire, how the column acceptance of medication in table 6 was derived. It seems like “acceptance” yes or no is a value related to the patient and not the medicine he/she takes.

This makes it difficult to understand the conclusions made such as the score lines are better accepted than tablets without. Might be an effect of reimbursement (prescribing a higher dose strengths) and asking the patient to split medication. In this context you should discuss dosing errors etc.

In the discussion you mention limitations of medications in Ethiopia, e,g. Not available in a different format (dispersible, with score line, …). Are those medications available in a different format in other countries / elsewhere?

Looking at the questionnaire itself it would have offered other important insights into dosing frequency, easy to take (with/without food),...Unfortunately those aspects are not mentioned in the article.

7. PLOS authors have the option to publish the peer review history of their article (what does this mean?). If published, this will include your full peer review and any attached files.

Reviewer #1: No

Reviewer #2: No

---

## [Author Response · Author response to Decision Letter 2]

13 Jan 2026

q. Introduction

Please be more specific in the introduction which aspects were assessed in the questionnaire and summarized under the term acceptability.

Please also explain more clearly why you expect that the results in Ethiopia differ from those reported for Europe.

R: Thank you for your constructive comments. We have revised the Introduction accordingly and addressed all the issues raised. In particular, we clarified the specific dimensions of medication acceptability assessed in this study.

Regarding the second comment, the rationale for expecting differences between findings from Ethiopia and those reported in European studies was strengthened based on the recommendations of Shariff et al. (2020). This study highlighted that existing European research does not adequately represent Black populations and may not fully capture the influence of sociocultural, healthcare system, and formulation-related factors relevant to low-resource settings. Our study was therefore designed in line with these recommendations to generate context-specific evidence and address this important gap in the literature.

q;What exactly do you mean by “Eligible participants …, based on patient discharge order.”?

R;

It is also unclear what you mean by “...those who held some of his/her medicine, and those who

were volunteers,...”? Thank you for pointing out this ambiguity. By “those who held some of his/her medicine,” we meant elderly patients who were able to physically present at least one of their currently prescribed oral solid medications at the time of the interview, allowing direct assessment of medication characteristics. “Those who were volunteers” refers to patients who voluntarily agreed to participate after being informed

q; What do you mean by “A maximum of 5 oral solid medications were taken from the elderly patients, which can cover more than 90 percent of their medications.”?

Thank you for requesting clarification. By this statement, we meant that for each Thank you for highlighting this point. By “eligible participants were selected … based on patient discharge order,” we meant that systematic random sampling was applied using the outpatient service flow. After patients completed their clinical consultation and received their medications from the hospital pharmacy, every Kth eligible elderly patient was approached for participation according to the order in which patients exited the outpatient department. This wording has been revised in the Methods section to improve clarity.participant, up to five currently used oral solid medications were selected for assessment. This limit was applied to reduce participant burden and interview duration, while still capturing the majority of medications used by elderly patients. Previous evidence indicates that evaluating up to five medications per patient accounts for more than 90% of chronically used medicines among elderly populations. The Methods section has been revised to clarify this point.

q;Have you checked that those variables, which you call independent, are not correlated to each other? For instance, education, income and residence or certain symptoms and diseases. Yes, potential correlations among independent variables were assessed during the analysis. Conceptually related variables (e.g., education, income, occupation and others) were first reviewed for overlap, and statistical multicollinearity was evaluated using variance inflation factors (VIFs) in the regression models. All variables included in the final multivariable analysis had VIF values within acceptable limits, indicating no problematic multicollinearity. This clarification has been added to the Methods section.

q;The terminology used in the text here, e.g. palatability is not used in the questionnaire itself. Did you use taste and other aspects to come up with a palatability score? If yes, how was this done? Thank you for this observation. The term palatability was used in the manuscript as a conceptual descriptor rather than as a direct questionnaire item. In the questionnaire, palatability-related aspects were assessed using specific items such as “…tastes good” and “….has no bad aftertaste.” These items were analyzed individually and, where appropriate, combined as part of the overall medication acceptability score derived from the MAQ-2019, rather than as a separate standalone palatability score.

q;Results and discussion

Figure 1: How representative is your study population compared to the population in Ethiopia in the same age range? Thank you for this comment. The study population was drawn from public tertiary and general hospitals in Addis Ababa, which serve as national referral centers and attract elderly patients from both urban and rural areas across Ethiopia due to the availability of better diagnostic and treatment services. Importantly, although the study was conducted in Addis Ababa, only 34.1% of the study participants were residents of Addis Ababa, while the majority originated from other regions of the country. This reflects the national referral role of these hospitals and enhances the geographic diversity of the study population. As a result, the age and sex distribution of participants is broadly comparable to the national older population, and the sample captures a wide range of chronic conditions and medication-use experiences from different parts of Ethiopia. While we acknowledge that the study is facility-based and may not represent elderly individuals who do not access hospital services, we believe it provides a reasonable representation of healthcare-utilizing elderly populations at the national level.

q; The terms in the questionnaire are often not clearly defined, such as swallowability difficulties, visual impairment, …Furthermore, looking at the questionnaire and the results I wonder how you managed in multimorbid patients using polypharmacy to distinguish if the amount of medications are attributed purely to the disease. Please explain how you did this, e.g. “Among the disease conditions affecting elderly patients, hypertension emerged as the most common (43.1%) with 50.2% of oral solid medication usage, followed by other cardiac-related disorders and diabetes mellitus.” First, key terms used in the analysis were operationalized through specific questionnaire items rather than as single, abstract concepts. For example, swallowability difficulties were assessed using multiple patient-reported items, including whether the medication: (i) feels stuck in the throat, (ii) requires excessive effort or repeated swallowing, (iii) causes choking or coughing during swallowing, and (iv) requires additional water or medication modification (e.g., crushing or splitting) to swallow. Visual impairment was similarly assessed using self-reported difficulty in reading medication labels, distinguishing tablets, or identifying score lines without assistance. Second, regarding multimorbidity and polypharmacy, we did not attempt to attribute individual medications to specific disease indications. Instead, patients’ chronic medical conditions (e.g., hypertension, diabetes, cardiac disorders) were identified from medical chart reviews, while information on medications was obtained from patient interviews and direct inspection of medicines. The reported result that 43.1% of patients were hypertensive and that these patients accounted for 50.2% of all oral solid medications used indicates that hypertensive patients many of whom also had additional chronic conditions were using a disproportionately higher number of medications overall. Importantly, this does not imply that 50.2% of medications were prescribed specifically for hypertension.

Each patient contributed all of their currently used oral solid medications to the analysis, regardless of indication. This approach reflects real-world medication use in elderly patients with multimorbidity and polypharmacy. We have revised the Methods and Results sections to clarify this point and prevent misinterpretation.

q; How was the taste assessed? How reliable is the result communicated based on that?

Table 5 mentioned “bitter color” as taste and “partial score line”. What is meant here? Thank you for raising this point. Taste was assessed using patient-reported perceptions rather than objective sensory testing. Specifically, participants were asked to rate statements such as “the medicine tastes good” and “the medicine has no bad aftertaste” using a Likert-scale format. These questions captured the patient’s subjective experience at the time of medication intake. While taste perception is inherently subjective, patient-reported assessment is considered appropriate and reliable for evaluating medication acceptability, particularly in elderly populations, as it reflects real-world use. The internal consistency of the acceptability-related items, including taste-related questions, was acceptable (Cronbach’s α = 0.77), supporting the reliability of the reported results.

Regarding Table 5, the term “bitter color” was a wording error and did not refer to taste. It has been corrected to “bitter taste” in the revised manuscript. The term “partial score line” refers to tablets with a score line that does not fully divide the tablet into equal halves or is not intended to ensure dose accuracy but primarily facilitates breaking for ease of swallowing. Both terms have now been clearly defined in the Methods and Operational Definitions sections to avoid confusion.

q; How was the judgement on the labelling performed? Was language taken into account? Labeling assessment took language into account; however, we clarify that none of the medications were labeled in Amharic. Judgement of labeling was therefore based on the patient’s ability to read and understand English-language labels, including instructions, font size, and clarity of information. Patients who reported being able to read and understand English labels had better labeling acceptance, whereas those with limited or no English proficiency reported greater difficulty.

q; From a methodological point of view I have some issues understanding based on the questionnaire, how the column acceptance of medication in table 6 was derived. It seems like “acceptance” yes or no is a value related to the patient and not the medicine he/she takes.

This makes it difficult to understand the conclusions made such as the score lines are better accepted than tablets without. Might be an effect of reimbursement (prescribing a higher dose strengths) and asking the patient to split medication. In this context you should discuss dosing errors etc.

Medication acceptance was assessed at the medication level, not solely at the patient level. Although the questionnaire was administered to patients, acceptance responses were collected separately for each oral solid medication currently used by the patient, up to a maximum of five medications per participant. For each medication, the MAQ-2019 items were answered with reference to that specific medicine (e.g., ease of swallowing, handling, taste, appearance, labeling, and willingness to take without modification). A composite acceptability score was calculated for each medication, and this score was dichotomized into “accepted” versus “not accepted” using the predefined threshold. Therefore, the acceptance outcome presented in Table 6 represents medication-level acceptability, while accounting for clustering of multiple medications within individual patients. This has been clarified in the Methods section.

We agree that acceptance of score-lined tablets may be influenced by prescribing practices, including higher dose strengths and the need for tablet splitting. However, in our study, the presence of a score line was evaluated from the patient perspective, specifically whether it facilitated easier handling or swallowing and reduced the need for uncontrolled breaking. We did not evaluate reimbursement-related prescribing decisions, as these are not applicable in the Ethiopian public healthcare context where medicines are largely subsidized or provided at low cost. We acknowledge that tablet splitting may introduce dosing inaccuracies, and this potential risk has now been explicitly discussed in the Discussion section as an important consideration when interpreting the higher acceptance of score-lined tablets.

q; In the discussion you mention limitations of medications in Ethiopia, e,g. Not available in a different format (dispersible, with score line, …). Are those medications available in a different format in other countries / elsewhere?

Regarding the availability of alternative formulations, such as dispersible tablets or age-adapted dosage forms, these formulations are commonly available in high-income countries and are often recommended by regulatory agencies (e.g., EMA) as part of patient-centered medicine design for elderly populations. In contrast, such alternatives are limited or largely unavailable in Ethiopia, where the market is dominated by standard generic solid dosage forms. We have revised the Discussion section to explicitly contrast the Ethiopian context with international availability and regulatory practices.

q; Looking at the questionnaire itself it would have offered other important insights into dosing frequency, easy to take (with/without food),...Unfortunately those aspects are not mentioned in the article. We acknowledge that the questionnaire included additional domains such as dosing frequency and ease of administration in relation to food. These variables were collected primarily as contextual and descriptive information and were not the primary focus of the current analysis, which was specifically designed to investigate physical formulation-related characteristics of oral solid medications. To maintain focus and avoid overextension of the manuscript, these aspects were not included in the final analysis. This has now been acknowledged as a limitation in the Discussion section.

---

## [Decision Letter · Decision Letter 2]

8 Feb 2026

The Need for Patient-Centric Medicine Design: Investigating Key Physical Characteristics of Oral Solid Medications to Improve Acceptance of Elderly Patients in Addis Ababa, Ethiopia.

PONE-D-25-43693R2

Dear Dr. Assefa,

We’re pleased to inform you that your manuscript has been judged scientifically suitable for publication and will be formally accepted for publication once it meets all outstanding technical requirements.

Kind regards,

James Mockridge

Staff Editor

PLOS One

Additional Editor Comments (optional):

Title and abstract in the submission system: Although you have updated the manuscript to change ‘elderly’ to ‘older’ in response to the reviewer's comments, please ensure that you have also updated the title and abstract in the submission system accordingly. Currently 'elderly' is still being used and this needs to be changed to 'older'.

Reviewers' comments:

Reviewer's Responses to Questions

**Comments to the Author**

1. If the authors have adequately addressed your comments raised in a previous round of review and you feel that this manuscript is now acceptable for publication, you may indicate that here to bypass the “Comments to the Author” section, enter your conflict of interest statement in the “Confidential to Editor” section, and submit your "Accept" recommendation.

Reviewer #1: All comments have been addressed

Reviewer #2: All comments have been addressed

2. Is the manuscript technically sound, and do the data support the conclusions?

Reviewer #1: Yes

Reviewer #2: Yes

3. Has the statistical analysis been performed appropriately and rigorously?

Reviewer #1: Yes

Reviewer #2: I Don't Know

4. Have the authors made all data underlying the findings in their manuscript fully available?

Reviewer #1: Yes

Reviewer #2: Yes

5. Is the manuscript presented in an intelligible fashion and written in standard English?

Reviewer #1: Yes

Reviewer #2: Yes

6. Review Comments to the Author

Reviewer #1: (No Response)

Reviewer #2: Dear authors,

thanks for up-dating the manuscript and also taking care for avoiding the term "elderly". It would be nice of this term could be replaced in the title as well.

Thanks

7. PLOS authors have the option to publish the peer review history of their article (what does this mean?). If published, this will include your full peer review and any attached files.

Reviewer #1: No

Reviewer #2: No

---

## [Editor Report · Acceptance letter]

PONE-D-25-43693R2

PLOS One

Dear Dr. Assefa,

I'm pleased to inform you that your manuscript has been deemed suitable for publication in PLOS One. Congratulations! Your manuscript is now being handed over to our production team.

Kind regards,

on behalf of

Dr James Mockridge

Staff Editor

PLOS One